# Two-Dimensional Position Tracking Using Gradient Magnetic Fields

**DOI:** 10.3390/s22145459

**Published:** 2022-07-21

**Authors:** Xuan Thang Trinh, Jen-Tzong Jeng, Huu-Thang Nguyen, Van Su Luong, Chih-Cheng Lu

**Affiliations:** 1Faculty of Mechanical Engineering, Hung Yen University of Technology and Education, Hungyen 160000, Vietnam; xttrinh@utehy.edu.vn; 2Department of Mechanical Engineering, National Kaohsiung University of Science and Technology, Kaohsiung 807618, Taiwan; i108142110@nkust.edu.tw; 3Faculty of Electrical and Electronic Engineering, Phenikaa University, Hanoi 12116, Vietnam; su.luongvan@phenikaa-uni.edu.vn; 4Institute of Mechatronics Engineering, National Taipei University of Technology, Taipei 106344, Taiwan; cclu23@ntut.edu.tw

**Keywords:** position tracking, induction coil, gradient magnetic fields, magnetic tracking

## Abstract

In this work, a two-dimensional (2D) position-detection device using a single axis magnetic sensor combined with orthogonal gradient coils was designed and fabricated. The sensors used were an induction coil and a GMR spin-valve sensor GF807 from Sensitec Inc. The field profiles generated by the two orthogonal gradient coils were analyzed numerically to achieve the maximum linear range, which corresponded to the detection area of the tracking system. The two coils were driven by 1-kHz sine wave currents with a 90° phase difference to generate the fields with uniform gradients along the *x*- and *y*-axis in the plane of the tracking stage. The gradient fields were detected by a single-axis sensor incorporated with a digital dual-phase lock-in detector to retrieve the position information. A linearity correction algorithm was used to improve the location accuracy and to extend the linear range for position sensing. The mean positioning error was found to be 0.417 mm, corresponding to the relative error of 0.21% in the working range of 200 mm × 200 mm, indicating that the proposed tracking system is promising for applications requiring accurate control of the two-dimensional position.

## 1. Introduction

Nowadays, magnetic sensors are the key devices in multiple applications; e.g., material characterization, position detections, rotation measurements, and motion tracking. Among them, motion-tracking techniques based on the detection of AC or pulsed DC fields are widely applied in human-body motion tracking [1], vehicle tracking [2], medical marking [3,4], etc. Several types of magnetic sensors have been successfully applied in tracking systems, such as the giant magnetoresistance (GMR) [5,6], anisotropic magnetoresistance (AMR) [7,8], Hall sensor [9,10], and fluxgate types [11,12]. The typical magnetic tracking techniques rely on the detection of the vector components of magnetic fields emitted by three-axis dipole transmitters [13,14]. Since the distribution of the dipole field is non-linear in space, a high-speed computational system is required to perform real-time digital signal processing to retrieve the position of the target sensor. An alternative method of magnetic tracking is to use a biaxial sensor in combination with a triad three-axial transmitter [15]. The distance resolution of this method is better than that of the conventional triad three-axial tracking method. Moreover, by using a large 2D array of uniaxial transmitters, the signal-to-noise ratio (SNR) and distance resolution, which are two important parameters in a position-tracking system, have been significantly improved [16]. Additionally, the optimization of the transmitting coil’s geometry also improves the dipole field approximation. It is promising that the combination of magnetic and inertial sensors for arbitrary position and orientation can compensate for the cumulative orientation error caused in dynamic motion. For example, the real-time six degrees of freedom (DOF) tracking of human motion was demonstrated [1,17]. However, in retrieving the position, the accuracy is limited by the distortion of magnetic fields caused by interference in the environment. Hence, to enhance the accuracy of magnetic tracking, the error-correction algorithm must be included to prevent the negative influence of magnetic field distortion [18,19]. To simplify the error correction algorithm, a feasible way is to design a magnetic field following a simple distribution function, e.g., the constant gradient magnetic field. For magnetic field mapping applications [6], the determination of the actual field point taken by the sensor is crucial to increasing the accuracy of the magnetic inverse problem [20]. Using the magnetic sensor to track its position itself would be a novel solution to solve this problem.

In this paper, we proposed a two-dimensional magnetic tracking method using a single-axis magnetic sensor as a target in a gradient field. Two kinds of magnetic sensors, including induction coil and GMR spin-valve sensor, are used to demonstrate a two-dimensional position-tracking system using gradient field in real-time. The gradient coil was designed, fabricated, and analyzed to achieve a field range with a stable constant gradient, allowing the direct conversion of the magnetic field into position with a single axis magnetic sensor. The results of two-dimensional magnetic tracking were evaluated, and the algorithm for further linearity correction was discussed.

## 2. Experimental Designs

### 2.1. Design of Two-Dimensional Magnetic Tracking System

The two-dimensional magnetic tracking system consists of a single-axis magnetic sensor and two sets of magnetic field gradient coils, as shown in Figure 1. A single-axis magnetic sensor, which is an induction coil or a GMR spin-valve sensor, is used as the target in a two-axis gradient field. The excitation waveform for synchronous detection is provided by a function generator, model AFG-2225 from GW-Instek. The in-phase and quadrature excitation signals are amplified by a two-channel power amplifier to drive the gradient coils. The output signal of the sensor is analyzed by a lock-in amplifier, Model 7270 DSP from the Signal Recovery, to resolve the in-phase and quadrature components. The analog outputs of the lock-in amplifier are connected to a data acquisition (DAQ) device, model USB-6216 from the National Instruments. The graphical user interface (GUI) software for signal recording and real-time processing is coded in LabVIEW. To investigate the spatial response of the system, the gradient coils are mounted on an X-Y translation stage while the sensor is fixed above the center of the detection plane with a fixed gap, as shown in Figure 1a. Two step-motors controlled by a PLC driver are used for driving the translation stage to move the gradient coil relative to the sensor.

The gradient magnetic field for position tracking is generated by the *x*- and *y*-gradient coils, as shown in Figure 2. The two gradient coils have the same size and coil turns. Each gradient coil consists of four elements, including two main coils and two shim coils. The element coils are formed by wrapping 200 turns of enameled copper wires around rectangular frames made of non-magnetic acrylic plastic material. The shim coils are 300 mm × 50 mm × 3 mm in dimension, and the main coils are 300 mm × 115 mm × 3 mm in dimension. The four-element coils are mounted on an acrylic substrate plate. The shim coils for the *x*-gradient field are fixed on the top of the plate, while the main coils are at the bottom. Moreover, the shim coils and main coils of the *y*-gradient coil are reversely arranged to the *x*-gradient coil. The four-element coils for each axis are connected in series, and the winding directions are arranged to generate the constant gradient field at the center of the tracking stage. The two gradient coils are mounted 20 mm apart and aligned orthogonally to each other.

The in-phase and quadrature-phase sinusoidal excitation currents *I_i_* and *I_q_* are injected into the *x*- and *y*-gradient coils, respectively. They are at the same frequency with the phases differed by 90°. The magnitude and polarity of gradient fields *B_i_* and *B_q_* are controlled by setting the amplitude and phase of the excitation current. The excitation signals from the function generator are amplified by an in-house-built power amplifier circuit to drive the coils with 1-kHz sine wave currents.

When the sensor moves on the detection plane above the gradient coils, its sensing direction is perpendicular to the plane. For the *x*-gradient coil, the distances from the detection plane to the shim and main coils are 66 mm and 71.5 mm, respectively.

For the *y*-gradient coil, the distances are 50.5 mm and 45 mm for the shim and main coils, respectively. The optimal distances between the element coils are determined by numerical simulation to maximize the linear range of field distribution. *B_i_* and *B_q_*, the *z*-axis field components for the *x*- and *y*-gradient coils, are calculated by calculating the contributions of the element coils, as follows:(1)$$B_{i}=B_{z,s}(+x)+B_{z,m}(+x)+B_{z,s}(-x)+B_{z,m}(-x),$$
(2)$$B_{q}=B_{z,s}(+y)+B_{z,m}(+y)+B_{z,s}(-y)+B_{z,m}(-y).$$
where the subscript “*s*” represents the shim coil and “*m*” indicates the main coil. The *z*-axis field component induced by each element coil is calculated by calculating the contribution of four current segments. For example, the field component for the +*x* shim coil is:(3)$$B_{z,s}(+x)=B_{z,s}(x_{1})+B_{z,s}(x_{2})+B_{z,s}(y_{1})+B_{z,s}(y_{2}),$$
where *B_z_*_,s_(*x*_1_), *B_z_*_,s_(*x*_2_), *B_z_*_,s_(*y*_1_), and *B_z_*_,s_(*y*_2_) are the contributions of AB, CD, BC and DA current segments, respectively, as shown in Figure 3. For each segment, the *z*-axis field component was calculated by the equation derived from the Biot–Savart law:(4)$$B_{z,s}(x_{1})=\frac{\mu_{0}I}{4\pi R}(\sin\alpha_{2}-\sin\alpha_{1})\cos\theta$$
in which
(5)$$R=\sqrt{{\left(x_{1}-x_{0}\right)}^{2}+z_{0}^{2}}$$
(6)$$\sin\alpha_{1}=\frac{\left(y_{2}-y_{0}\right)}{\sqrt{{\left(x_{1}-x_{0}\right)}^{2}+{\left(y_{2}-y_{0}\right)}^{2}+z_{0}^{2}}}$$
(7)$$\sin\alpha_{2}=\frac{\left(y_{1}-y_{0}\right)}{\sqrt{{\left(x_{1}-x_{0}\right)}^{2}+{\left(y_{1}-y_{0}\right)}^{2}+z_{0}^{2}}}$$
and
(8)$$\cos\theta =\frac{\left(x_{1}-x_{0}\right)}{\sqrt{{\left(x_{1}-x_{0}\right)}^{2}+z_{0}^{2}}}$$

The contribution of the other element coils, i.e., *B_z_*_,s_(*x*_2_), *B_z_*_,s_(*y*_1_), and *B_z_*_,s_(*y*_2_), were calculated in the similar way. The obtained field distributions for the *x*- and *y*-gradient coils on the detection plane are shown in Figure 4. It was found that *B_i_* and *B_q_* increase monotonically along the axis of the gradient. The field *B_i_* of *x*-coil is almost linear in the range of *x* = ±0.1 m, while the field *B_q_* of *y*-coil exhibits more nonlinearity. The shorter distance from the detection plane to the *y*-coil causes deterioration in the uniformity of the gradient of *B_q_* but enhances the mean-field gradient. When a linearity-correction algorithm is applied, the usable dynamic range of position tracking for the *y*-coil becomes larger than that for the *x*-coil.

The field profiles generated by the two orthogonal gradient coils were analyzed numerically to achieve the maximum linear range, which corresponded to the detection area of the tracking system. The simulation of field distributions was performed using the ANSYS software with the excitation current of 4.28 mA. The simulated field distributions shown in Figure 5. Figure 5a,b illustrate the vector field distribution of *x*- and *y*-coil, respectively. From Figure 4 and Figure 5c,d, it can be observed that the calculated and simulated field distributions of *B_i_* and *B_q_* are similar in terms of the curve shape and amplitude.

To verify the simulation results, the field distribution generated by the combined gradient coils driven by a 1-kHz current was measured by an induction coil, as shown in Figure 6. The working range on the *x*–*y* plane is 200 mm × 200 mm in dimension and the sampling points are spaced by 20 mm for both the *x*- and *y*-directions. The field distribution on the detection plane was obtained by recording the in-phase and quadrature-phase outputs from the sensor output, i.e., *V_i_* and *V_q_*_,_ respectively, using a dual-phase lock-in amplifier. It was found that the field distribution of *x*-coil was more linear than that of the *y*-coil, in agreement with the simulation results. The simulation results showed that, in the working range, the maximum amplitude was about 4.2 µT for the *B_q_* (*x*-gradient) field and more than 6.3 µT for the *B_q_* (*y*-gradient) field with the excitation current of 4.28 mA. The measured field distribution is shown in Figure 6, where the maximum field amplitude in the working range is about 4.18 µT for *B_i_* and 6.8 µT for *B_q_*. The minor discrepancy in *B_q_* values between the simulated and experimental results may be attributed to the gain errors in the amplifying circuits for generating the driving currents in the *x*- and *y*-coils. The field distribution was also verified with the coils driven by a direct current using a commercial fluxgate sensor. Since the variation in the size, shape, and alignment of the coil may also induce the error in field magnitude, the comprehensive method for making the correction is experimental calibration.

### 2.2. Working Principle

The position of the target is obtained by detecting the gradient fields *B_i_* and *B_q_* generated by the two gradient magnetic coils, of which the coordinates are correlated to the field distribution. The sensor output, which contains the mixed signals from the *x*- and *y*-gradient coils, is analyzed by a digital dual-phase lock-in amplifier with a total gain of 50. The use of the phase-sensitive detection technique avoids interference from environmental disturbance, and hence optimizes the signal-to-noise ratios (SNR) [21,22]. The process for retrieving the position signals with dual-phase synchronous detection is illustrated in Figure 7. The waveform of sensor output is multiplied by the in-phase and quadrature-phase reference signals synchronized to the excitation current with a phase difference φ. When φ is tuned to the correct value, the dual-phase outputs *V_i_* and *V_q_* correspond to the *x-* and *y-*coordinates of the target magnetic sensor, respectively. The response time of the system is determined by the low-pass filter with a time constant of 10 ms. The dual-phase outputs of the lock-in amplifier, *V_i_* and *V_q_*, are recorded by a 16-bit DAQ controlled by a GUI software. The linearity correction algorithm is performed in real time to retrieve the position of the sensor on the detection plane.

Let *B_i_* and *B_q_* be the in-phase and quadrature-phase amplitudes of the *z*-axis component generated by the planar gradient coils, as shown in Figure 2. In the linear range, they have constant gradients along the *x*- and *y*- axes, respectively. In the ideal case, their intensities are linear functions of the spatial coordinates *x* and *y*. When the origin of an *x*-*y* plane is set to be at the center of the gradient coils, *B_i_* and *B_q_* are proportional to *x* and *y* coordinates, respectively:(9)$$B_{i}=g_{x}x\;$$
(10)$$B_{q}=g_{y}y\;$$
where *gx* = *dB_i_*/d*x* and *gy* = *dB_q_*/d*y* are the constant gradients of *B_i_* and *B_q_*. In our system, the waveforms of *B_i_* and *B_q_* are sinusoidal at the same frequency (1 kHz), but the phases differ by 90°. When a magnetic field sensor detects the *z*-axis component *B*_z_ at a field point on the same *x-y* plane, it can be represented by *B_i_* and *B_q_*_,_ as follows:(11)$$B_{z}(\omega t)=B_{i}\cos(\omega t)+B_{q}\sin(\omega t)=B_{0}\cos(\omega t-\phi )$$
where *B*_0_ = (*B_i_*^2^ + *B_q_*^2^)^1/2^ is the amplitude and *ϕ* = atan(*B_q_*/*B_i_*) is the phase of the *z*-axis component. In terms of *B*_0_ and *ϕ*, *B_i_* = *B*_0_ cosϕ and *B_q_* = *B*_0_ sinϕ. The spatial coordinates *x* and *y* can be resolved from the sensor outputs using a dual-phase synchronous detection with a phase reference to the excitation currents in the coils. Given the field-to-voltage transfer coefficient *k* for the field sensor, the orthogonal output voltages, *V_x_* and *V_y_*, correspond to the fields as follows:(12)$$V_{x}=kB_{i}=kg_{x}x\;$$
(13)$$V_{y}=kB_{q}=kg_{y}y$$

In general, the output of the detection circuit has a reference phase difference *φ* for *B_z_*. With a non-zero *φ*, the corresponding in-phase and quadrature-phase outputs *V_i_* and *V_q_* are both functions of *φ*:(14)$$V_{z}(\varphi )=k\frac{2}{T}\int_{0}^{T}B_{z}(\omega t)\cos(\omega t+\varphi )dt=kB_{0}\cos(\phi +\varphi )\;$$
where *ϕ* = arctan(*V_y_*/*V_x_*). With the phase differences *φ* and φ−90°, the in-phase and quadrature-phase outputs are associated with *V_x_* and *V_y,_* as follows: (15)$$V_{i}=V_{z}(\varphi )=V_{x}\cos\varphi -V_{y}\sin\varphi \;$$
(16)$$V_{q}=V_{z}(\varphi -90^{\circ })=V_{x}\sin\varphi -V_{y}\cos\varphi$$

This can be expressed in matrix representation:(17)$$\left[\begin{array}{c} V_{i}\\ V_{q} \end{array}\right]=\left[\begin{array}{cc} \cos\varphi & -\sin\varphi \\ \sin\varphi & \cos\varphi \end{array}\right]\left[\begin{array}{c} V_{x}\\ V_{y} \end{array}\right]$$

Therefore, the coordinates *V_x_* and *V_y_* on the voltage plane are:(18)$$\left[\begin{array}{c} V_{x}\\ V_{y} \end{array}\right]=\left[\begin{array}{cc} \cos\varphi & \sin\varphi \\ -\sin\varphi & \cos\varphi \end{array}\right]\left[\begin{array}{c} V_{i}\\ V_{q} \end{array}\right]$$

According to (18), the change in reference phase *φ* corresponds to a rotation on the *V_x_*-*V_y_* voltage plane, which results in a rotation in the spatial coordinates *x* and *y* about the *z*-axis. By adjusting the phase angle φ, one may redefine the orientation of the coordinate axes, providing that the gradients are perfectly constant. Using Equations (12) and (13) and (18), the spatial coordinates *x* and *y* can be calculated from the output voltage as follows:(19)$$\left[\begin{array}{c} x\\ y \end{array}\right]=\frac{1}{kg_{x}g_{y}}\left[\begin{array}{cc} g_{y}\cos\varphi & g_{y}\sin\varphi \\ -g_{x}\sin\varphi & g_{x}\cos\varphi \end{array}\right]\left[\begin{array}{c} V_{i}\\ V_{q} \end{array}\right]$$

For the gradients |*g_x_*| = |*g_y_*|, the aspect ratio between the *x*- and *y*- axes is 1, which means that the coordinates are not distorted under a rotation induced by changing the reference phase angle φ.

When the excitation currents were applied to the gradient coils, no crosstalk between the coils was observed since the currents were delivered by independent power amplifiers. As the phases of applied currents in the gradient coils were orthogonal to each other, the distortion in position tracking was dominated by the non-uniform field gradient, which can be corrected by the proposed algorithm.

### 2.3. The Linearity Correction Method

To correct the remaining non-uniformity of gradient field distribution, the linearity correction algorithm was applied to suppress the influence of distorted magnetic fields to improve the accuracy of position tracking. As an initial guess of the sensor’s position, the measured magnetic field was directly converted into the position based on the sensor outputs of in-phase *V_i_* and quadrature-phase *V_q_* using Equation (19). The linearity correction algorithm was performed by applying the cubic spline interpolation method to obtain smooth contour curves of the implicit function *B_z_*(*x*,*y*) = constant for both in-phase and quadrature-phase components. To implement the linearity correction algorithm, the databases of *V_i_* and *V_q_* were collected by stepping the sensor on the tracking stage with 121 sampling points. The sampling points were spaced by 20 mm for both the *x*- and *y*- directions. The *x*- and *y*- databases were expanded by using a cubic spline interpolation for the two-dimensional field distribution. Hence, the size of the interpolated databases was extended to 441 data points for *x*- and *y*- gradient magnetic fields. The interpolated field distributions of *x*- and *y*- coils consist of twenty-one natural cubic spline functions for *y*- and *x*- directions, respectively, as shown in Figure 6. The interpolated curves of *x*- and *y*- gradient field distribution have a smoother profile, similar to the theoretically predicted distributions. Therefore, the gradient field distribution can be precisely mapped with the limited number of sampled field points.

The cubic spline functions for the *x*- and *y*-gradient field distributions are *B_i_*(*x, y_m_*) and *B_q_*(*x_m_, y*), respectively. The in-phase interpolating function *B_i_*(*x, y_m_*) consists of the different cubic functions *B_i_**_n_*(*x*) in twenty equally spaced intervals from *x*_1_ to *x*_21_:(20)$$B_{i}(x,y_{m})=\left\{\begin{array}{cc} B_{i1}(x) & x_{1}\leq x\leq x_{2}\\ B_{i2}(x) & x_{2}\leq x\leq x_{3}\\ \vdots & \vdots \\ B_{i20}(x) & x_{20}\leq x\leq x_{21} \end{array}\right.$$
where *B_in_*(*x*)= *a_in_* + *b_in_x* + *c_in_x*^2^ + *d_in_x*^3^ (*d_in_* ≠ 0) is a natural cubic function with *n* = 1, 2, 3,..., 21. The function *B_i_*(*x, y_m_*) passes through *B_i_*(*x_n_*) = *B_in_*, where *n* = 1, 3, 5,…, 21 are the measured data points and *n* = 2, 4, 6,…, 20 are the interpolated data points. The coordinate *y_m_* corresponds to the *m*’th grid line, *m* = 1, 2,..., 21, of the *x*-gradient field distribution. Similarly, the quadrature-phase cubic spline interpolating function *B_q_*(*x_m_*, *y*) along *y*-direction is:(21)$$B_{q}(x_{m},y)=\left\{\begin{array}{cc} B_{q1}(y) & y_{1}\leq y\leq y_{2}\\ B_{q2}(y) & y_{2}\leq y\leq y_{3}\\ \vdots & \vdots \\ B_{q20}(y) & y_{20}\leq y\leq y_{21} \end{array}\right.$$
where *B_qn_*(*y*) = *a_qn_* + *b_qn_y* + *c_qn_y*^2^ + *d_qn_y*^3^ is a natural cubic function with *n* = 1, 2, 3,..., 21. The function *B_q_*(*x_m_*, *y*) passes through *B_q_*(*x_n_*) = *B_qn_*, where *n* = 1, 3, 5,…, 21 are the measured data points and *n* = 2, 4, 6,…, 20 are the interpolated data points. The coordinate *x_m_* corresponds to the *m*’th grid line, *m* = 1, 2,..., 21, of the *y*-gradient field distribution. Suppose that the field components detected by the magnetic sensor at (*x*_0_, *y*_0_) are (*B_i_*_0_, *B_q_*_0_), and the in-phase and quadrature-phase components of sensor output are *V_i_*_0_ and *V_q_*_0_. The sensor position (*x*_0_, *y*_0_) can be retrieved by performing linear interpolations to calculate the intersection of two contour lines, *B_i_*(*x*,*y*) = *B_i_*_0_ and *B_q_*(*x*,*y*) = *B_q_*_0_ using the extended databases, as shown in Figure 8.

At the sensor position (*x*_0_, *y*_0_), the corresponding field components are (*B_i_*_0_, *B_q_*_0_). To implement the linearity correction algorithm, the two-dimensional cubic spline interpolation was applied by using the inverse interpolating function *x*(*B_i_, y_m_*) and *y*(*B_q_*, *x_m_*). Since *B_i_* and *B_q_* increase monotonically along the axis of the gradient, the field components are compared to the values of the *B_i_* and *B_q_* interpolated databases, respectively, to find the two contour lines *B_i_*(*x,y*) = *B_i_*_0_ and *B_q_*(*x,y*) = *B_q_*_0_. For instance, to find the contour line *B_i_*(*x,y*) = *B_i_*_0_, comparison of *B_i_*_0_ value to the *B_i_*− interpolated database is performed. For each *y_m_* value (*m* = 1, 2, 3,..., 21), an interval (*B_in_*_−1_, *B_in_*) is found, satisfying a condition *B_in_*_−1_ ≤ *B_i_*_0_ ≤ *B_in_*. The (*B_in_*_−1_, *B_in_*) intervals corresponding to the (*x_n_*_−1_, *x_n_*) interval satisfying a condition *x_n_*_−1_ ≤ *x*_0_ ≤ *x_n_* (*n* = 2, 3,...,21) are the landmarks in the database. Therefore, a natural cubic spline function in (*x_n_*_−1_, *x_n_*) interval is determined using Equation (20). The coordinates (*x*_0*m*_, *y_m_*) corresponding to *B_i_*_0_ value is given by implementing the cubic spline interpolation with inverse function *x*(*B_i_, y_m_*). The points (*x*_0*m*_, *y_m_*) forming a contour line of *x*− interpolated database are *B_i_*(*x*,*y*) = *B_i_*_0_. Similarly, the obtained points (*x_m_*, *y*_0*m*_) forming a contour line of *y*-interpolated database are *B_q_*(*x*,*y*) = *B_q_*_0_.

The position of the sensor is determined by finding an intersection of two contour lines. To find out the intersection point of two contour lines, the four nearest points around the intersection are determined by comparing the distance between the fixed interval points on *x*- and *y*- contour lines. The two *y*-contour points closest to the *x*-contour line are identified by obtaining two minimum distances, and the two nearest points on the *x*- and *y*- contour lines are identified in the same way. The four obtained points form two intersecting segments, of which the intersection point (*x*_0_, *y*_0_) gives the sensor position.

## 3. Results and Discussion

As the proposed system converts the magnetic field directly into position based on the dual-phase synchronous detection technique, the signal processing is straightforward and fast. The sensor’s output signals are almost a linear function of *x*- and *y*-coordinates. The working range, which is 200 mm × 200 mm in dimension, is slightly less than the coil area. When moving the induction coil sensor on the detection plane, the coordinates of the sensor are displayed in real-time on the GUI software by performing the linearity correction algorithm. To evaluate the feasibility and accuracy of the linearity correction algorithm, the sensor was scanned on the detection plane by moving the gradient coils with the X–Y translation stage. The coordinates of the sensor were recorded with an interval of 10 mm along the *x*- and *y*-directions in the working range of 200 mm × 200 mm. The retrieved coordinates of 441 data points are shown in Figure 9 and Figure 10a. Figure 9 shows that the coordinates along *x*- and *y*-axis are linear and uniform in gradient. In the comparison to the data without the linearity correction algorithm in Figure 6, the result in Figure 9 shows that the non-linearity component is well-suppressed by the linearity correction algorithm based on the cubic spline interpolation method.

Figure 10a shows the position deviation between the set and retrieved coordinates for the 441 sensor positions retrieved in real time with a grid size of 1 cm × 1 cm. The distance between the set position and the retrieved position was calculated to estimate the positioning error of each retrieved position on the working range. The positioning error along *x*- and *y*- axes at each data point was calculated using the following formula:(22)$$E_{n}=\sqrt{E_{xn}^{2}+E_{yn}^{2}}$$
where *E_n_* is positioning error at the data point *n*, and *E_xn_* and *E_yn_* are the positioning error at the data point *n* along the *x*- and *y*-axes, respectively. The average positioning error is obtained by taking the average value of 441 positioning errors.

The positioning errors of the position-tracking system are shown in Figure 10b. It was found that the data points around the center of the working range had lower positioning errors while the positioning errors of the marginal points were higher. The average positioning error was found to be 0.417 mm, corresponding to the relative error of 0.21% in the working range of 200 mm × 200 mm. In Figure 10a, for the central region within |x| ≤ 5 and |y| ≤ 5, the average positioning error is as small as 0.309 mm (the red area), corresponding to the positioning error of 0.15% in the working range. For the locations near the margins at |x| > 8 and |y| > 8 (the blue area), the average positioning error is 0.458 mm (0.27%). The maximum error of 1.612 mm (0.81%) occurs at the coordinate of (9,9). The displacement error induced by the root-mean-square noise of the induction coil sensor are 0.34 and 0.21 mm for displacement along the x- and y-axis, respectively. The results show that the linearity correction algorithm can overcome the distortion caused by the non-uniform field gradient and determine the sensor’s two-dimensional position with high precision and accuracy in real-time.

The positioning accuracy achieved in our work is generally better than the existing magnetic-field-based methods reported for three- or two-dimensional position tracking [23,24,25]. For instance, the magnetic positioning of the capsule endoscope using an array of 16 three-axis Hall sensors and a magnetic tube is reported in [23], in which the random complex algorithm (RCA) gave better results than the Levenberg–Marquardt algorithm (LMA). The positioning error achieved was less than 8.34 mm in the working range of 240 mm × 240 mm, corresponding to an error of 3.5%. This system was applied in a gastrointestinal tract capsule endoscope of the human body. In [25], a Hall effect sensor array, which consists of 25 Hall effect sensors, was used to estimate the position of the permanent magnet in capsule endoscope. The experimental results show that the root mean square error (RMSE) for the estimated position value of the permanent magnet was less than 1.13 mm in the X, Y, and Z directions. Another example is the position-tracking system using magnetic landmarks for autonomous mobile robots with the estimation method using NVM (Neutral Magnetic Valley) [24]. For this method, the positioning errors are generally less than 3 mm in the 130 mm range of a single cell. The proposed sequential landmark search technique allows the mobile robot to move with respect to the magnetic landmark’s geometric center from an unknown initial position. However, the relative error of 2.3% is still high. In comparison with the existing techniques, our method can achieve an average positioning error of 0.21% (corresponding to 0.417 mm), which is more promising for high accuracy applications of position tracking.

The existence of additional non-magnetic and non-conductive objects in the sensing range does not affect the positioning accuracy. However, the placement of magnetic and/or conductive objects in the sensing area does induce positioning error. Specifically, the interference to the amplitude and phase of the gradient field induces a DC offset in the output of the lock-in amplifier. When the additional object is fixed in its shape, orientation, and position, the DC offset voltage can be eliminated by the linearity correction algorithm.

To better define the target position, the system was also tested with a spin-valve GMR sensor, model GF708 from Sensitec GmbH, to replace the induction coil. Figure 11 shows the result of real-time 2D position tracking using the GMR sensor. The positioning errors near the center and margin of the working region were found to be 0.21% and 0.44%, respectively, and the average positioning error was found to be 0.35%. The observed errors were higher than those with the induction coil sensor, the indicating that the induction coil sensor is more accurate than the GMR spin-valve sensor in position tracking. Nevertheless, as the feature size of the GMR sensor is only 1.4 mm, which is much smaller than the 10 mm radius of the coil, it is more suitable for tracking a miniature moving object on a 2D plane. The limitation imposed by the finite sensor size can be relaxed by scaling up the gradient coil. In this way, the system can be made suitable for large-scale 2D positioning application. Further work to add more sensors to determine the height and orientation of the target will make the proposed technique useful for three-dimensional position tracking.

## 4. Conclusions

The design and performance of a two-dimensional position-tracking system using an induction coil to sense the orthogonal gradient magnetic fields were implemented and verified. With the orthogonal gradient fields being 90° out of phase, the synchronously detected phase angle corresponds to the rotation in the two-dimensional spatial coordinate. The influence of non-uniformity in the gradient field is suppressed by employing the linearity correction algorithm based on the cubic spline interpolation method. The accuracy in positioning is enhanced by applying the linearity correction algorithm. The achieved average positioning error is 0.21% in the 200 mm × 200 m working range and 0.15% in the 100 mm × 100 mm central region. The results show that the interpolation-based linearity correction algorithm is feasible in improving the precision and accuracy of real-time two-dimensional magnetic tracking. The proposed tracking device is promising for medical and industrial applications requiring accurate position control of the magnetic sensor, such as two-dimensional m–agnetic field mapping and magnetic source imaging [26].

## Figures and Tables

**Figure 1 sensors-22-05459-f001:**
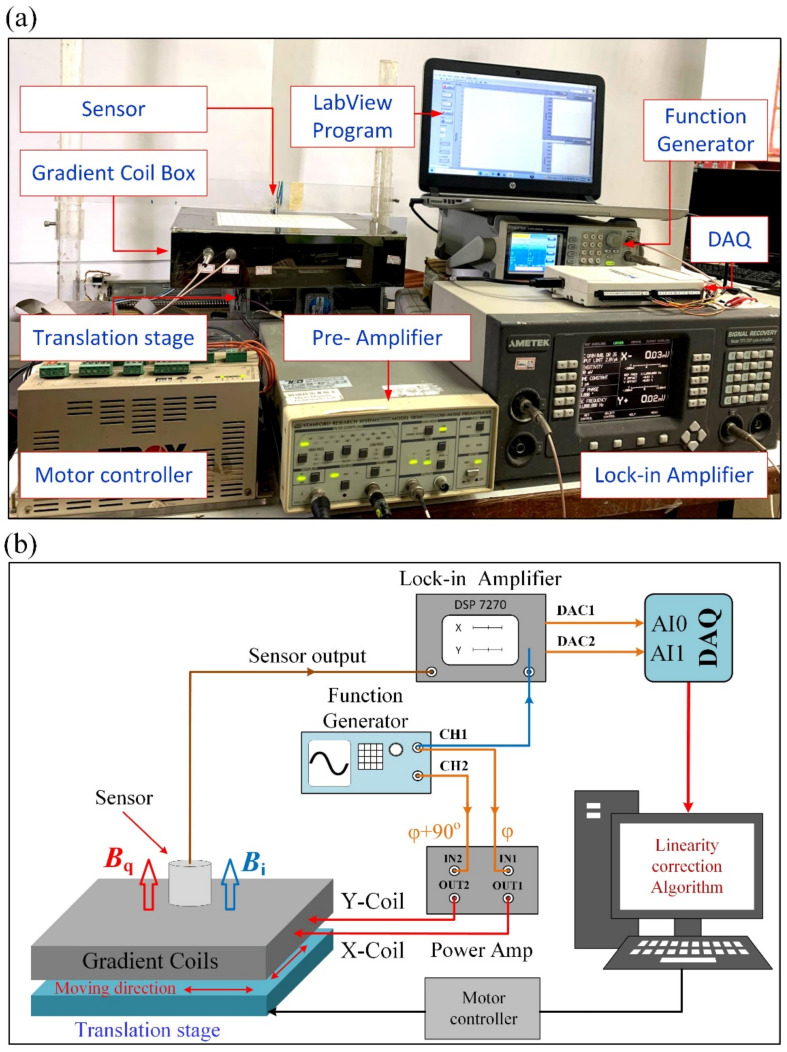
Schematic diagram of the 2D magnetic position tracking system. (**a**) Photograph of the system, and (**b**) Function block diagram.

**Figure 2 sensors-22-05459-f002:**
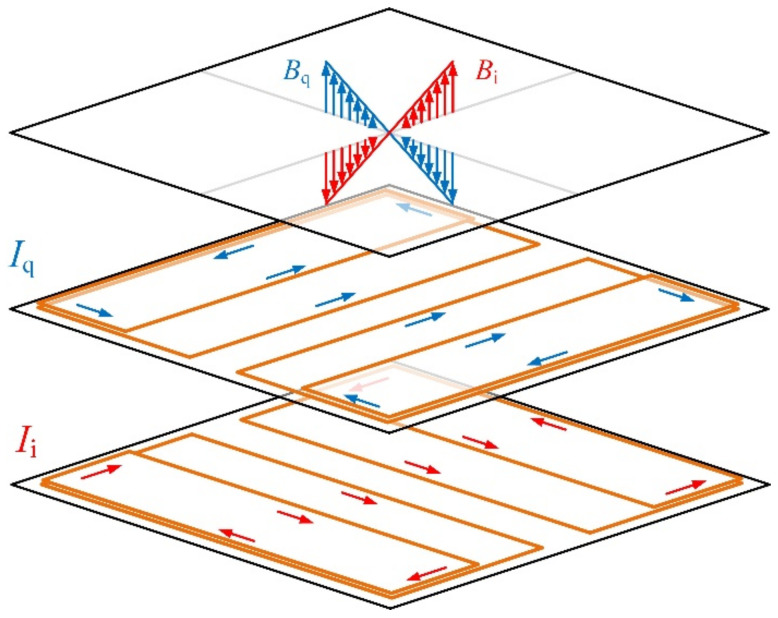
The layout of *x*- and *y*-gradient coils for constant gradient fields.

**Figure 3 sensors-22-05459-f003:**
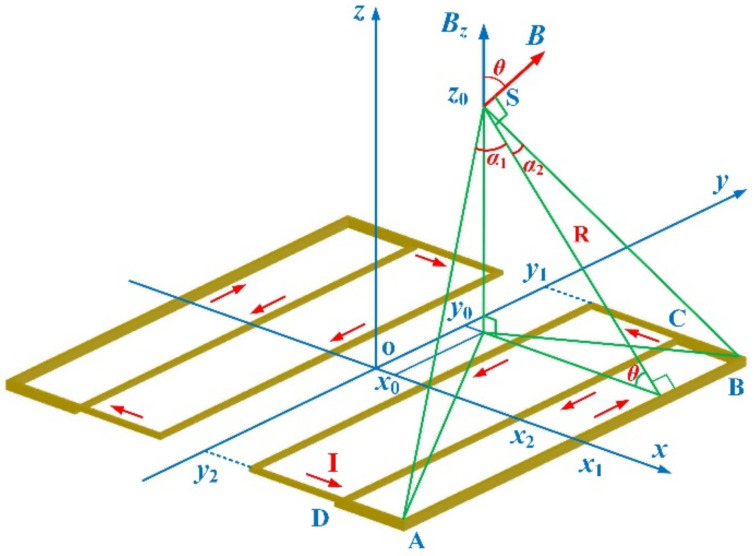
The field component of the shim coil generated by the AB (*x*_1_) segment.

**Figure 4 sensors-22-05459-f004:**
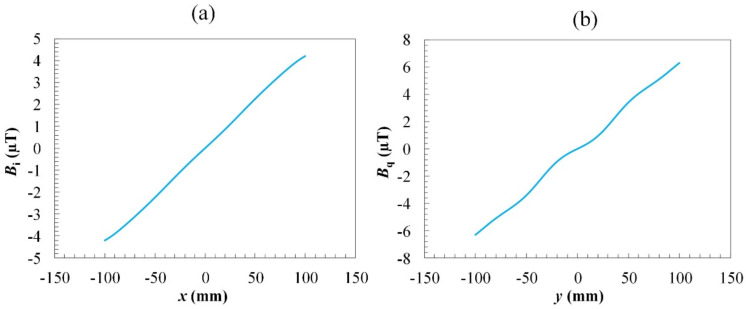
The calculated field distribution with the excitation current of 4.28 mA of (**a**) *x*-coil (*B_i_*) at *y* = 0 and (**b**) *y*-coil (*B_q_*) at *x* = 0.

**Figure 5 sensors-22-05459-f005:**
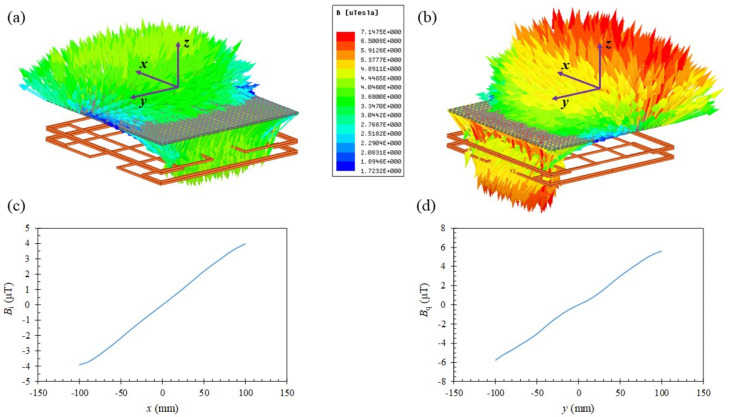
The simulated gradient magnetic field distribution on detection plane of (**a**) *x*-coil (*B_i_*), (**b**) *y*-coil (*B_q_*), (**c**) *B_i_* at *y* = 0, and (**d**) *B_q_*at *x* = 0.

**Figure 6 sensors-22-05459-f006:**
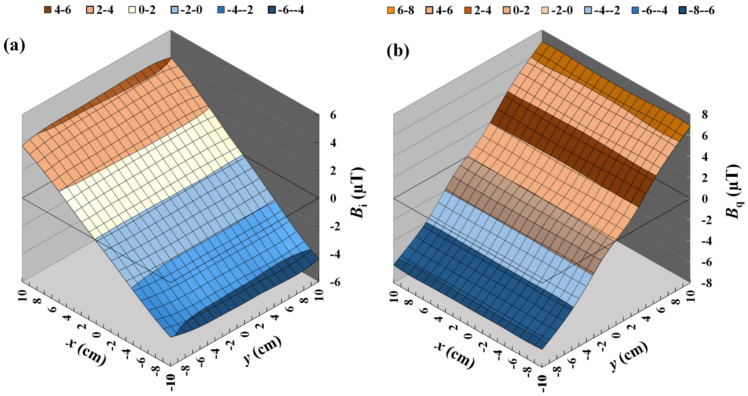
The measured field distributions of (**a**) *x*–coil (*B_i_*) and (**b**) *y*–coil (*B_q_*). The total number of field points is 11 × 11 = 121 and the grid size is 2 cm × 2 cm. A cubic spline interpolation was used to obtain the smooth distribution.

**Figure 7 sensors-22-05459-f007:**
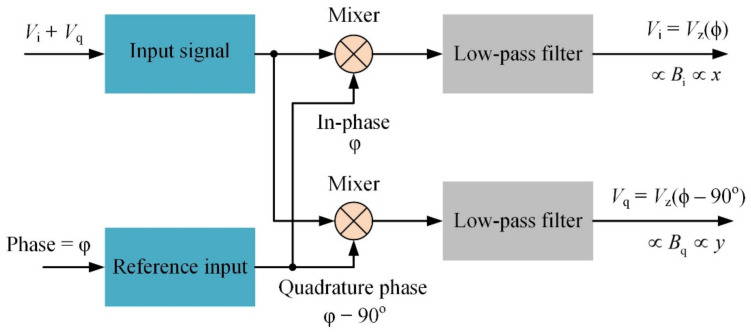
Process for retrieving the position signals with dual-phase synchronous detection.

**Figure 8 sensors-22-05459-f008:**
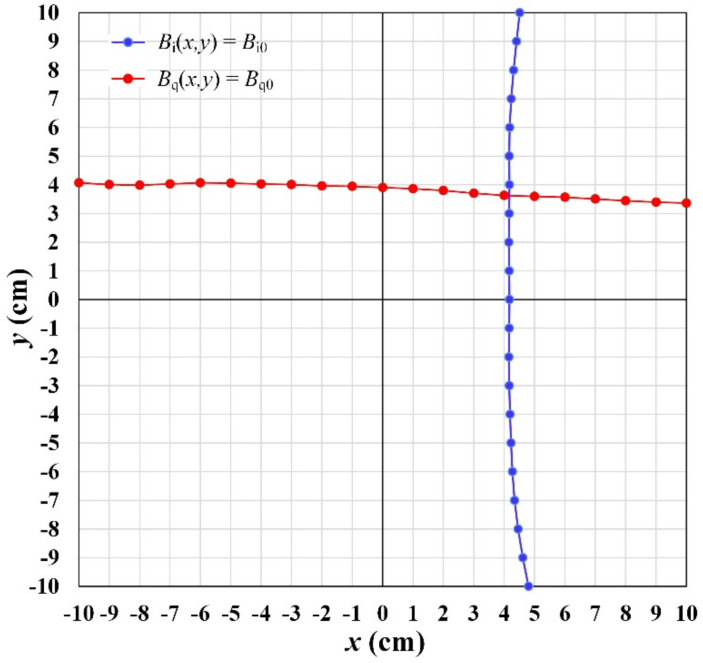
The two contour lines corresponding to *B_i_*_0_ and *B_q_*_0_ plotted by using the cubic interpolation algorithm. The sensor position is determined by the intersection of two contour lines.

**Figure 9 sensors-22-05459-f009:**
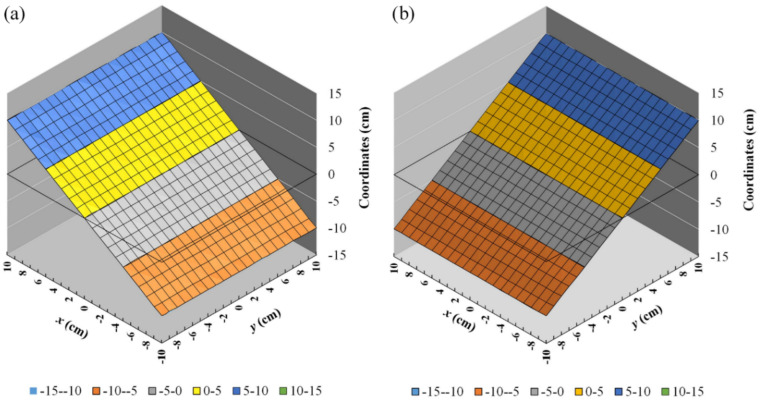
The coordinates converted from the field distribution along (**a**) *x*-axis and (**b**) *y*-axis after applying the linearity correction algorithm. The total number of data points is 21 × 21 = 441, and the grid size is 1 cm×1 cm.

**Figure 10 sensors-22-05459-f010:**
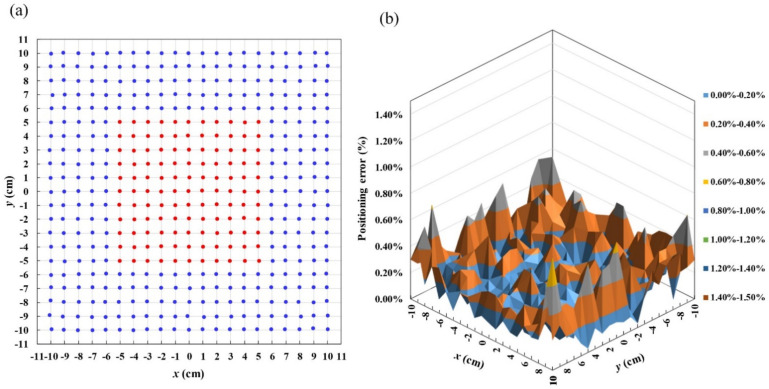
Retrieved coordinates and positioning errors of the 2D position-tracking system using an induction coil. (**a**) Retrieved 441 sensor positions in real time with grid size of 1 cm × 1 cm and (**b**) positioning error of tracking system.

**Figure 11 sensors-22-05459-f011:**
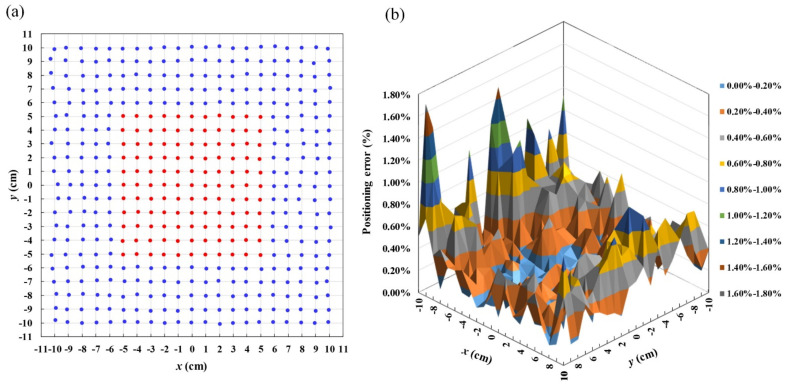
Retrieved coordinates and positioning errors of the 2D position-tracking system using GMR spin-valve sensor. (**a**) Retrieved 441 sensor positions in real time with grid size of 1 cm × 1 cm and (**b**) positioning error of tracking system.

## Data Availability

Not applicable.

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
