# Peer review of "Two-Dimensional Position Tracking Using Gradient Magnetic Fields"

_sensors, 2022, doi:10.3390/s22145459_

Round 1

Reviewer 1 Report

The authors designed a two-dimensional position tracking system using an induction coil by sensing the orthogonal gradient magnetic fields.

The background the well presented the motivation is well described. The implementation section clearly described the hardware platform, the signal processing pipeline, and the final result with different sensors. 

A minor revision is however needed, and a few questions:

1, Is there crosstalk between the x and y gradient coils when supplying the current? If yes, do the equations used in the paper for deriving the field information still correctly describe the truth? 

2, Where is Figure 3? 

3, Line 162: The minor discrepancy in Bq values between the simulated and experimental results may be attributed to the gain errors in the amplifying circuits for generating the driving currents in the x- and y-coils. Is it possible to record to driving current value to track the gain error and confirm the source of the minor discrepancy? 

4, The color labels in Figures 6 and 9 seem not to match the figures. 

Author Response

1.       Is there crosstalk between the x and y gradient coils when supplying the current? If yes, do the equations used in the paper for deriving the field information still correctly describe the truth?

Answer: Thank you for raising the important question. When the excitation currents are applied to the gradient coils, no crosstalk between the coils was observed since the currents are delivered by independent power amplifiers. As the phases of applied currents in the gradient coils are orthogonal to each other, the distortion in position tracking is dominated by the non-uniform field gradient, which can be corrected by the proposed algorithm. The description related to this issue is added to the paragraph before Sec. 2.3 in the revised manuscript.

2.       Where is Figure 3?

Answer: Thank you very much for pointing out this problem. Figure 3 was lost in preparing the submission version of manuscript. It is recovered in the revised manuscript.  

3.       Line 162: The minor discrepancy in Bq values between the simulated and experimental results may be attributed to the gain errors in the amplifying circuits for generating the driving currents in the x- and y-coils. Is it possible to record to driving current value to track the gain error and confirm the source of the minor discrepancy?

Answer: It is possible to minimize the discrepancy by monitoring the phase and amplitude of the driving current and making the correction. However, since the size, shape, and alignment error of the coil may also alter the field magnitude, the comprehensive method for making the correction is the experimental calibration. The description related to this issue is added to the paragraph before Sec. 2.2 in the revised manuscript.

4.       The color labels in Figures 6 and 9 seem not to match the figures.

Answer: Thank you very much for pointing out this issue. We used the 3D model to illustrate in Figure 6 and 9, in which the colors in the legends are made the same. The colors on the curved surfaces in Figures 6 and 9 seem not to match because the viewing angles of two figures are different. The shading of the image affects the color and makes it different from the legend label. This problem has been corrected by changing the color in the legend to make it better match the curved surface.  

Reviewer 2 Report

"Moreover, by using a large array of 2-D, the signal-to-noise ratio (SNR) and distance resolution, ..."

There is something missing in the sentence, a large array of what?

The quality of equations is very low, all equations should be rewritten. 

The quality of figures 9, 10 and 11 should be improved. 

When field distributions are presented, authors use smooth distribution achieved by cubic spline interpolation. However, it is not clear what real data look like. Such information could be quite interesting for readers.

Moreover, it is not clear how the data was collected, i.e. how many measurements of field distribution were taken at each point.

Authors present achieved average localization error, however, it is unclear how many position estimates were performed. 

The authors claim that the average positioning error is 0.417mm. Could the authors provide some insight on how was the ground truth position estimated? What was the uncertainty of the ground truth position? 

Can authors provide a more detailed comparison to other positioning methods based on magnetic field measurements? 

What is the potential application of the proposed solution, how will the placement of other objects in the area affect the positioning accuracy of the proposed system? 

Is it possible to scale up the system to provide positioning service in large-scale deployment scenarios?

Reviewer 3 Report

Work analyzed position tracking of a single-axis magnetic sensor in gradient magnetic field.

Tracking precision has been experimentally estimated for a particular situation when sensing axis is strictly orthogonal to the plane of gradient coils and does not change this orientation during the movement.

Im not sure it could be a case in any practical application. Moreover, the more distance from the center of the coil system, the more value of x- and y- components of the coils field that can be picked-up by the sensor declined from precisely vertical orientation. In this case, precision of the position tracking could be, probably, degraded significantly. I suggest the authors will describe, at least briefly, how the presented method will be awaited to work in practical applications.

Line 120. I have not found Fig.3.

Line 172-173.   gradient fields, with the magnetic..  This sentence should be re-formulated.

Fig. 10 and Fig. 11. As far as Ive understood, you have placed a contour map over a 3D map. However, the last one is obscured by a contour map plane. I suggest placing the contour map separately and instead of Fig. 10a and 11a which, in turn, do not add much to readers understanding.

Author Response

Reviewer 3:

Work analyzed position tracking of a single-axis magnetic sensor in gradient magnetic field.

Tracking precision has been experimentally estimated for a particular situation when sensing axis is strictly orthogonal to the plane of gradient coils and does not change this orientation during the movement.

1.       Line 120. I have not found Fig.3.

Answer: Sorry for the error. Figure 3 has been added to the revised manuscript.

2.       Line 172-173. “  gradient fields”, “ with the magnetic..”  This sentence should be re-formulated

Answer: We have revised this sentence as “by the two gradient magnetic coils” at Line 177 in the revised manuscript.

3.       Fig. 10 and Fig. 11. As far as I’ve understood, you have placed a contour map over a 3D map. However, the last one is obscured by a contour map plane. I suggest placing the contour map separately and instead of Fig. 10a and 11a which, in turn, do not add much to reader’s understanding.

Answer: Thank you very much for your valuable comment. Fig. 10b and 11b are re-plotted and replaced in the revised version. As to Fig.10a and Fig.11a, our intentions are to show the collected position data points with the correction algorithm and the range of positioning errors.
